# Impact of prognostic nutritional index on outcomes in patients with *Mycobacterium avium* complex pulmonary disease

**Sung Woo Moon**[1], **Eun Hye Lee**[1], **Ji Soo Choi**[1], **Ah Young Leem**[1], **Su Hwan Lee**[1], **Sang Hoon Lee**[1], **Song Yee Kim**[1], **Kyung Soo Chung**[1], **Ji Ye Jung**[1], **Moo Suk Park**[1], **Young Sam Kim**[1], **Young Ae Kang**[1,2]*

**1** Division of Pulmonology, Department of Internal Medicine, Severance Hospital, Yonsei University College of Medicine, Seoul, Republic of Korea, **2** Institute of Immunology and Immunological Diseases, Yonsei University College of Medicine, Seoul, Republic of Korea

\* mdkang@yuhs.ac

## Abstract

Onodera's prognostic nutritional index (PNI) is useful in predicting prognosis of various diseases. But the usefulness of PNI in non-surgical patients has not been sufficiently proven yet. In patients with mycobacterium avium complex pulmonary disease (MAC-PD), malnutrition is an important factor that affects the quality of life and morbidity. Here, we aimed to evaluate whether PNI is related with clinical outcomes in MAC-PD patients. We examined 663 patients diagnosed with MAC-PD between May 2005 and November 2017. PNI score was calculated at the time of diagnosis and treatment initiation, and patients were divided into malnutrition and non-malnutrition groups according to a cut-off PNI score of 45. As the recommended duration of treatment for MAC-PD is 12 months following sputum conversion, treatment duration less than 12 months was defined as treatment intolerance. Survivals were compared with the log-rank test. Multivariate logistic regression and multivariate Cox proportional hazards models were used to estimate the odds ratio (OR) and hazards ratio (HR) for treatment intolerance and mortality, respectively. Of the 306 patients that received treatment, 193 received treatment longer than 12 months. In the multivariable logistic regression model, malnutrition at the time of treatment initiation was related with treatment intolerance (OR: 2.559, 95% confidence interval [CI]: 1.414–4.634, P = 0.002). Patients in the malnutrition group at the time of diagnosis exhibited lower survival (P<0.001) and malnutrition at the time of diagnosis was a significant risk for all-cause mortality (HR: 2.755, 95% CI: 1.610–4.475, P<0.001). Malnutrition, as defined by PNI, is an independent predictor for treatment intolerance and all-cause mortality in patients with MAC-PD.

## Introduction

Malnutrition is generally associated with immune dysfunction and inflammatory processes, [1] leading to diminished quality of life and increased mortality in patients with pulmonary disease. [2, 3] Patients with malnutrition and limited respiratory reserves often have quantitative and functional alterations in skeletal and respiratory muscles. [1, 4] Numerous indicators,

**Data Availability Statement:** All relevant data are within the manuscript and its Supporting Information file.

**Funding:** The authors received no specific funding for this work.

**Competing interests:** The authors have declared that no competing interests exist.

including body composition, serum protein, and nutritional indices, such as nutritional risk screening [5] and subjective global assessment, have been used as markers to reflect nutritional status. [6] Among the various nutritional assessment tools, Onodera's prognostic nutritional index (PNI) [7] assesses serum albumin levels and total lymphocyte counts in the peripheral blood. PNI was originally proposed to assess the perioperative nutritional status and surgical risk in patients undergoing gastrointestinal surgery. Studies have shown PNI to be a versatile prognostic tool for various malignancies. [8, 9] Compared to other nutritional indexes, PNI is an index easily calculated using only serum albumin levels and lymphocyte counts and is therefore easily assessable. But the usefulness of PNI in non-cancer medical patients has not been sufficiently proven yet.

Nontuberculous mycobacterial pulmonary disease (NTM-PD) is becoming an increasingly common diagnosis. [10–12] The most common etiology of NTM-PD is *Mycobacterium avium complex* pulmonary disease (MAC-PD), [13] which is associated with an impaired quality of life [14] and is difficult and costly to treat. [15] Recommended treatment regimens for MAC-PD include macrolides, ethambutol, and rifampin. The American Thoracic Society (ATS) guidelines recommend a 12-month treatment period for MAC-PD following sputum conversion; however, drug intolerance often limits successful therapy. [16] In one previous study, more than one third of MAC-PD patients discontinued their medication because of drug intolerance, and only 33% of MAC-PD patients who started treatment achieved culture conversion. [17]

Poor nutritional status indicated by various markers such as body mass index (BMI) and low body fat is known to be an important factor for the pathogenesis and prognosis of NTM-PD. [18–20] However, its clinical implications in the treatment of patients with MAC-PD have not been evaluated as a composite value.

Therefore, in this study, we aimed to evaluate whether patients with malnutrition, as defined by PNI, are at (1) higher odds of intolerance to treatment, (2) lower odds of achieving culture conversion after treatment than non-malnutrition patients, and (3) higher risk of mortality than non-malnutrition patients.

## Materials and methods

### Study design and population

This study was a retrospective cohort analysis conducted at a tertiary care hospital. The cohort included patients diagnosed with MAC-PD between May 2005 and November 2017 (Fig 1). Patients were selected from our retrospective NTM-PD registry based on the following criteria: MAC-PD was confirmed according to the criteria by ATS. [15] Initially, 861 patients diagnosed with MAC-PD were included in the study. Patients were excluded if: (1) computed tomography (CT) images at the time of diagnosis according to our institutional radiology database were not available (n = 74); (2) clinical data, including age, height, body weight, smoking history, laboratory results, and acid-fast bacilli (AFB) test results were unavailable (n = 114); (3) there was a prior diagnosis of MAC-PD (n = 2); (4) there was a history of lung transplantation (n = 4); or (5) they were infected by hepatitis B virus (n = 2) or human immunodeficiency virus (n = 2). In total, 663 patients were included in the analysis.

At the time of diagnosis, data on age, smoking history, laboratory test results, underlying diseases, height and weight, radiographic findings, AFB test results, and symptoms were collected from all patients. Age was categorized into two groups (age < 65 vs. ≥ 65 years). AFB test results were categorized into three groups (negative, '1+ and 2+', and '3+ and 4+'). Radiographic findings were categorized according to the presence of cavitary lesion on CT images.

After the diagnosis of MAC-PD, data regarding laboratory test results and antibiotic therapy were collected from 306 patients who underwent antibiotic therapy. Laboratory test results

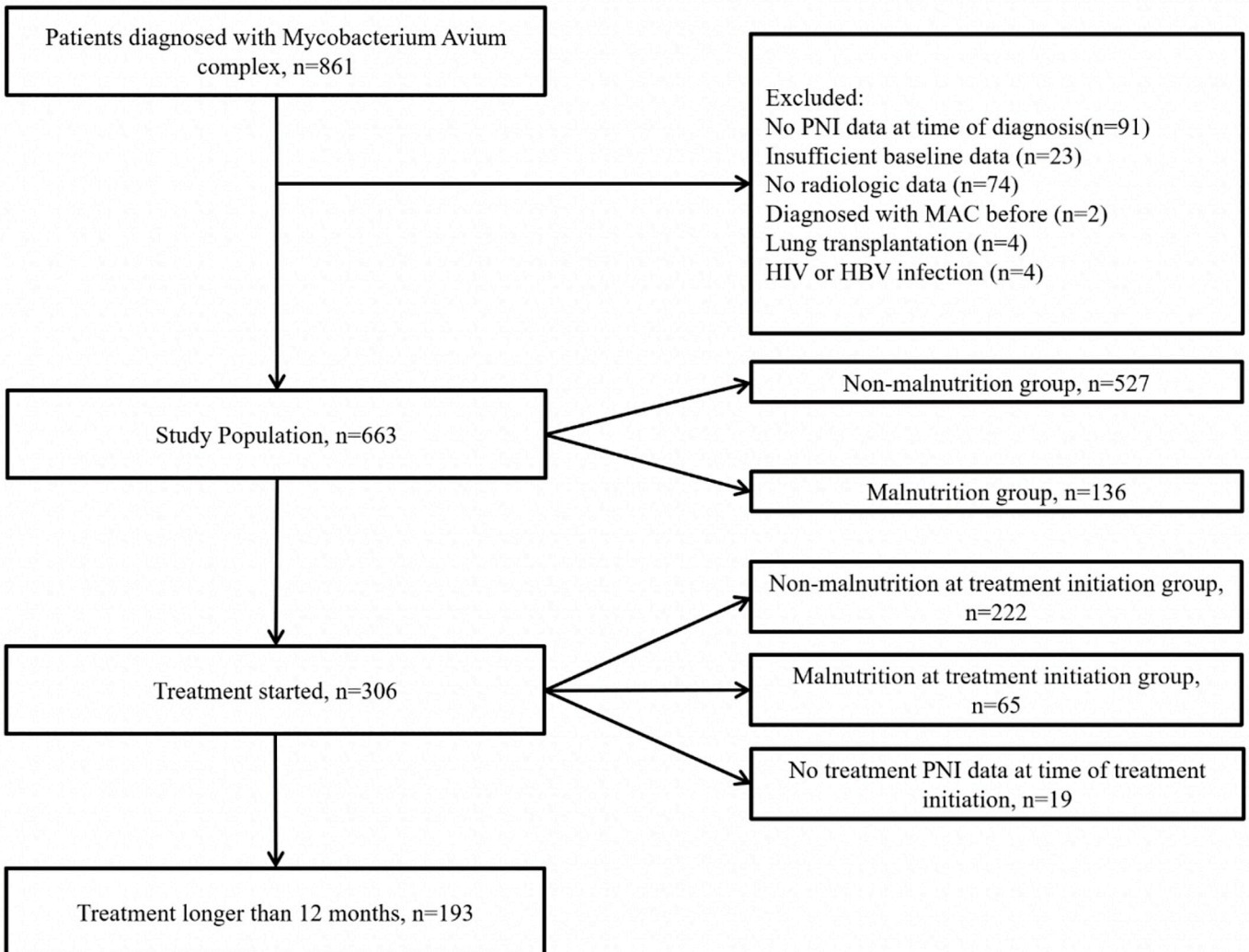

**Fig 1. Flow diagram of subjects in this study.** List of Abbreviations: HBV, Hepatitis B Virus; HIV, Human Immunodeficiency Virus; MAC, Mycobacterium avium complex; PNI, prognostic nutritional index.

at the time of treatment initiation were available from 278 patients, and culture conversion data were available from 177 patients among 193 patients who underwent antibiotic therapy for more than a year. As the recommended duration of treatment for MAC-PD is 12 months following sputum conversion, treatment duration less than 12 months was defined as treatment intolerance. Culture conversion was defined as the presence of at least three consecutive negative mycobacterial cultures from respiratory samples collected at least 4 weeks apart, in accord with the 2018 NTM-NET consensus statement [21]; Date of culture conversion was established based on sampling date of the first negative culture. Follow-up data including culture results and mortality were collected until October 2018. Mortality was estimated based on the date of diagnosis to death or the last follow-up. The primary outcome was all-cause mortality. Secondary outcomes were treatment intolerance and culture conversion after antibiotic treatment.

## Surrogate markers for evaluation of the nutritional status

The PNI score was calculated using the following formula at the time of MAC-PD diagnosis and treatment initiation: $10 \times$ serum albumin value (g/dL) + $0.005 \times$ total lymphocyte count in the peripheral blood (/mm$^3$). Patients were divided into malnutrition and non-malnutrition groups according to a cut-off PNI score of 45. [22–24]

## Statistical analysis

Chi-squared tests and student's t-tests were used to compare categorical and continuous variables, respectively, between the two groups. Multivariate logistic regression models with backward variable selection were used to estimate the odds ratios (ORs) for malnutrition and culture conversion while controlling potential confounding factors. Survival was estimated using the Kaplan–Meier method and compared using the log-rank test. Multivariate Cox proportional hazards models were used to investigate relationships between clinical parameters and mortality. Of the variables collected, serum albumin levels and lymphocyte counts were not included in the multiple logistic regression models because they were included in the calculation of PNI. Variables with a p-value (P) < 0.10, as determined by the log-rank test, were included in the multivariate Cox proportional hazard ratio model. Variables included in all multivariable analysis were tested for multicollinearity. An adjusted P < 0.05 was considered statistically significant. All statistical analyses were performed using SPSS version 25.0 (SPSS Inc., Chicago, IL, USA).

## Ethics statement

This research protocol was approved by the Institutional Review Board / Ethics committee. (Severance Hospital, Yonsei University Health System Institutional Review Board) The requirement to obtain informed patient consent was waived because of the retrospective nature of this study.

# Results

## Baseline characteristics

The baseline characteristics of participants with and without malnutrition at the time of diagnosis are shown in Table 1. Patients with malnutrition were older, predominantly male, and had a lower BMI. The malnutrition group also had a higher proportion of participants with a history of chronic kidney disease, cardiovascular, and malignancy than did the non-malnutrition group. Based on the PNI calculations, malnutrition group had lower PNIs than the non-malnutrition group (median: 38.3 vs. 50.0, P < 0.001). Those in the malnutrition group also had a higher proportion of cavitary CT findings and positive AFB smear results. Symptomatically, the malnutrition group reported more symptoms of dyspnea, fever, and general weakness. Of 663 patients with MAC-PD, 63 died during the follow-up period. Malnutrition patients showed significantly higher mortality rates during the follow-up (P < 0.001).

## Variables related to treatment intolerance

Among 663 MAC-PD patients, 306 (46.1%) patients were treated for MAC-PD with multiple antibiotics. To evaluate the factors related to treatment intolerance, these 306 patients were stratified by treatment duration as follows (Table 2): 113 (36.9%) patients received treatment for less than 12 months (shorter treatment group), and 193 (63.1%) patients received treatment for longer than 12 months (longer treatment group). One third of patients did not stick to the medication as scheduled.

**Table 1. Baseline clinical characteristics of 663 patients with Mycobacterium avium complex lung disease with or without malnutrition.**

| Variables | All (n = 663) | Non-malnutrition (n = 527) | Malnutrition (136) | P-value |
|---|---|---|---|---|
| Age, years | 64.1 ± 11.8 | 62.5 ± 11.6 | 70.4 ± 10.8 | <0.001 |
| Age ≥ 65 years | 343 (51.7%) | 243 (46.1%) | 100 (73.5%) | <0.001 |
| Gender, male | 292 (44.0%) | 206 (39.1%) | 86 (63.2%) | <0.001 |
| BMI, kg/m$^2$ | 20.5 ± 3.3 | 20.8 ± 3.1 | 19.7 ± 3.7 | <0.001 |
| Smoking, pack-years | 28.8 ± 23.2 | 28.5 ± 25.2 | 29.7 ± 15.8 | 0.787 |
| Comorbidities | | | | |
| Hypertension | 235 (35.4%) | 184 (34.9%) | 51 (37.5%) | 0.615 |
| Diabetes | 97 (14.6%) | 71 (13.5%) | 26 (19.1%) | 0.103 |
| Chronic liver disease | 43 (6.5%) | 30 (5.7%) | 13 (9.6%) | 0.117 |
| Chronic kidney disease | 33 (5.0%) | 22 (4.2%) | 11 (8.1%) | 0.075 |
| Cardiovascular disease | 93 (14.0%) | 62 (11.8%) | 31 (22.8%) | 0.002 |
| Malignancy | 170 (25.6%) | 130 (24.7%) | 40 (29.4%) | 0.271 |
| Laboratory test | | | | |
| PNI* at time of diagnosis | 50.0 ± 10.5 | 53.0 ± 9.3 | 38.3 ± 5.7 | <0.001 |
| Lymphocyte at time of diagnosis, 1000 cells/μL | 1.87 ± 1.59 | 2.08 ± 1.72 | 1.09 ± 0.46 | <0.001 |
| Albumin at time of diagnosis, g/dL | 4.1 ± 0.6 | 4.3 ± 0.4 | 3.3 ± 0.5 | <0.001 |
| PNI* at time of treatment initiation§ | 49.1 ± 6.6 | 51.0 ± 4.5 | 42.1 ± 7.9 | <0.001 |
| Cavitary lesion on computed tomography | 155 (23.4%) | 113 (21.4%) | 42 (30.9%) | 0.023 |
| AFB smear | | | | 0.011 |
| Negative | 569 (85.8%) | 463 (87.9%) | 106 (77.9%) | |
| 1+ or 2+ | 69 (10.4%) | 48 (9.1%) | 21 (15.4%) | |
| 3+ or 4+ | 25 (3.8%) | 16 (3.0%) | 9 (6.6%) | |
| Symptoms | | | | |
| Cough | 279 (42.1%) | 216 (41.0%) | 63 (46.3%) | 0.284 |
| Sputum | 288 (43.4%) | 231 (43.8%) | 57 (41.9%) | 0.699 |
| Dyspnea | 77 (11.6%) | 42 (8.0%) | 35 (25.7%) | <0.001 |
| Hemoptysis | 107 (16.1%) | 89 (16.9%) | 18 (13.2%) | 0.36 |
| Fever | 42 (6.3%) | 20 (3.8%) | 22 (16.2%) | <0.001 |
| Weakness | 35 (5.3%) | 11 (2.1%) | 12 (8.8%) | 0.001 |
| Treatment started | 306 (46.2%) | 241 (45.7%) | 65 (47.8%) | 0.700 |
| Treatment duration > 12 months | 193 (29.1%) | 165 (31.3%) | 28 (20.6%) | 0.015 |
| Time from diagnosis to initiation of treatment, months | 12.0 ± 21.3 | 13.1 ± 22.5 | 8.0 ± 15.3 | 0.096 |
| Follow-up period, months | 47.3 ± 33.5 | 50.1 ± 33.5 | 36.5 ± 31.3 | <0.001 |
| Culture conversion within a year after treatment‡ | 161 / 239 (67.4%) | 113 / 194 (68.6%) | 28 / 45 (62.2%) | 0.481 |
| Death during follow-up | 63 (9.5%) | 34 (6.5%) | 29 (21.3%) | <0.001 |

Data are presented as number (%) for categorical variables and median (range) or mean ± standard deviation (SD) for continuous variables

List of Abbreviations: PNI, prognostic nutritional index; BMI, body mass index; AFB, acid-fast bacilli

*PNI = 10 × serum albumin value (g/dL) + 0.005 × total lymphocyte count in the peripheral blood (/mm$^3$)

§PNI data at time of treatment initiation were available for 287 patients

‡Culture conversion data within 1 year after treatment were available for 239 patients

Most patients were treated with macrolides (97.4%), rifampin (97.7%), ethambutol (96.1%), and aminoglycosides (13.7%). Drugs such as isoniazid (2.9%) and fluoroquinolones (3.6%) were less frequently used. The mean treatment duration was 14.2 ± 10.1 months overall, with a mean duration of 5.1 ± 3.9 and 19.5 ± 8.7 months in the shorter and longer treatment groups, respectively. Patients in the shorter treatment group had lower PNI scores at either the time of

**Table 2. Basic characteristics of Mycobacterium avium complex lung disease patients who started treatment according to treatment tolerance (12 months).**

| Variables | All (n = 306) | Treatment longer than 12 months (n = 193, 63.1%) | Treatment shorter than 12 months (n = 113, 36.9%) | P-value |
|---|---|---|---|---|
| Age ≥ 65 years | 141 (46.1%) | 85 (44.0%) | 56 (49.6%) | 0.406 |
| Gender, male | 132 (43.1%) | 82 (42.5%) | 50 (44.2%) | 0.811 |
| BMI, kg/m$^2$ | 19.9 ± 3.3 | 19.8 ± 3.0 | 20.1 ± 3.8 | 0.433 |
| Smoking, pack-years | 34.4 ± 27.4 | 34.8 ± 25.7 | 33.9 ± 30.1 | 0.905 |
| Comorbidities | | | | |
| Hypertension | 92 (30.1%) | 54 (28.0%) | 38 (33.6%) | 0.305 |
| Diabetes | 44 (14.4%) | 26 (13.5%) | 18 (15.9%) | 0.613 |
| Chronic liver disease | 24 (7.8%) | 13 (6.7%) | 11 (9.7%) | 0.382 |
| Chronic kidney disease | 12 (3.9%) | 7 (3.6%) | 5 (4.4%) | 0.765 |
| Cardiovascular disease | 39 (12.7%) | 23 (11.9%) | 16 (14.2%) | 0.597 |
| Malignancy | 71 (23.2%) | 48 (24.9%) | 23 (20.4%) | 0.402 |
| Laboratory test | | | | |
| PNI* at time of diagnosis | 49.8 ± 9.0 | 51.7 ± 9.0 | 46.7 ± 8.0 | < 0.001 |
| Malnutrition at time of Diagnosis (PNI* < 45) | 67 (23.3%) | 30 (16.9%) | 37 (33.9%) | 0.001 |
| PNI* at time of treatment initiation$^\S$ | 49.1 ± 6.6 | 50.4 ± 5.7 | 46.9 ± 7.3 | < 0.001 |
| Malnutrition at time of treatment start$^\S$ (PNI < 45) | 65 / 287 (23.3%) | 30 (16.9%) | 37 (33.9%) | < 0.001 |
| Cavitary lesion on computed tomography | 112 (36.6%) | 76 (39.4%) | 36 (31.9%) | 0.219 |
| AFB smear | | | | 0.944 |
| Negative | 243 (79.4%) | 153 (79.3%) | 90 (79.6%) | |
| 1+ or 2+ | 45 (14.7%) | 28 (14.5%) | 17 (15.0%) | |
| 3+ or 4+ | 18 (5.9%) | 12 (6.2%) | 6 (5.3%) | |
| Symptoms | | | | |
| Cough | 135 (44.1%) | 80 (41.5%) | 55 (48.7%) | 0.234 |
| Sputum | 142 (46.4%) | 88 (45.6%) | 54 (47.8%) | 0.723 |
| Dyspnea | 41 (13.4%) | 20 (10.4%) | 21 (18.6%) | 0.055 |
| Hemoptysis | 59 (19.3%) | 44 (22.8%) | 15 (13.3%) | 0.051 |
| Fever | 19 (6.2%) | 10 (5.2%) | 9 (8.0%) | 0.337 |
| Weakness | 11 (3.6%) | 6 (3.1%) | 5 (4.4%) | 0.542 |
| Time from diagnosis to treatment, months | 12.2 ± 21.4 | 10.5 ± 18.9 | 14.9 ± 24.8 | 0.095 |
| Treatment medication | | | | |
| Macrolide | 298 (97.4%) | 187 (96.9%) | 111 (98.2%) | 0.715 |
| Rifampin | 299 (97.7%) | 189 (97.9%) | 110 (97.3%) | 0.712 |
| Ethambutol | 294 (96.1%) | 184 (95.3%) | 110 (97.3%) | 0.545 |
| Isoniazid | 9 (2.9%) | 5 (2.6%) | 4 (3.5%) | 0.730 |
| Fluoroquinolones | 11 (3.6%) | 6 (3.1%) | 5 (4.4%) | 0.542 |
| Aminoglycosides | 42 (13.7%) | 31 (16.1%) | 11 (9.7%) | 0.168 |
| Other | 9 (2.9%) | 7 (3.6%) | 2 (1.8%) | 0.493 |
| Initial number of medications | | | | 0.472 |
| ≤ 2 | 7 (2.3%) | 5 (2.6%) | 2 (1.8%) | |
| 3 | 252 (82.4%) | 155 (80.3%) | 97 (85.8%) | |
| ≥ 4 | 47 (15.4%) | 33 (17.1%) | 14 (12.4%) | |
| Duration of treatment, months | 14.2 ± 10.1 | 19.5 ± 8.7 | 5.1 ± 3.9 | < 0.001 |
| Culture conversion within a year after treatment$^\ddagger$ | 160 / 237 (67.5%) | 113 / 177 (63.8%) | 47 / 60 (78.3%) | 0.040 |

*(Continued)*

**Table 2.** (Continued)

| Variables | All (n = 306) | Treatment longer than 12 months (n = 193, 63.1%) | Treatment shorter than 12 months (n = 113, 36.9%) | P-value |
|---|---|---|---|---|
| Death during follow-up | 27 (8.8%) | 11 (5.7%) | 16 (14.2%) | 0.020 |

Data are presented as number (%) for categorical variables and median (range) or mean ± standard deviation (SD) for continuous variables

List of Abbreviations: AFB, acid-fast bacilli

*PNI = 10 × serum albumin value (g/dL) + 0.005 × total lymphocyte count in the peripheral blood (1000 cells/µL)

§PNI data at time of treatment initiation were available for 287 patients

‡Culture conversion data within 1 year after treatment were available for 237 patients

diagnosis or treatment initiation and showed higher culture conversion rates and mortality rates during the follow up.

Table 3 shows the relationship between malnutrition and treatment intolerance in the logistic regression models. When age (> 65 years), gender, symptoms of dyspnea, hemoptysis, and malnutrition were included in the regression model, malnutrition (OR: 2.559, 95% confidence interval [CI]: 1.414–4.634, P = 0.002) was significantly related to treatment intolerance.

## Variables related to culture conversion

To evaluate the factors related to culture conversion, patients who received treatment longer than a year and whose culture conversion data were available were stratified by culture conversion (Table 4). Among the 193 patients who received treatment longer than 12 months, culture conversion data were available for 177 patients. Among these 177 patients, culture conversion was achieved in 113 patients (63.8%). Patients who failed culture conversion were more likely to be male, have a shorter time span between diagnosis and treatment, have a lower BMI, and be treated longer. However, malnutrition at the time of diagnosis or treatment initiation did not significantly correlate with culture conversion.

S1 Table shows the relationship between malnutrition and culture conversion failure in the multivariate logistic regression models. When age, gender, cavitary lesion on CT, AFB smear result, time from diagnosis to treatment, and malnutrition at the time of treatment were included in the regression model, malnutrition (OR: 1.288, 95% CI: 0.630–2.637, P = 0.488) was not significantly related to culture conversion failure.

**Table 3. Multivariate logistic regression analyses for variables related to treatment intolerance.**

| Variables | OR (95% CI) | P-value |
|---|---|---|
| Age ≥ 65 years | 1.147 (0.677–1.942) | 0.611 |
| Gender, Male | 0.820 (0.475–1.414) | 0.475 |
| Dyspnea | 1.522 (0.738–3.139) | 0.255 |
| Hemoptysis | 0.557 (0.284–1.094) | 0.089 |
| Malnutrition, treatment initiation (PNI* <45) | 2.559 (1.414–4.634) | 0.002 |

List of Abbreviations: OR, odds ratio; 95% CI, 95% confidence interval; AFB, acid-fast bacilli; PNI, prognostic nutritional index

*PNI = 10 × serum albumin value (g/dL) + 0.005 × total lymphocyte count in the peripheral blood (1000 cells/µL)

**Table 4. Basic characteristics of Mycobacterium avium complex lung disease patients who received treatment longer than 12 months according to conversion failure**[*].

| Variables | All (n = 177) | Conversion achieved (n = 113) | Conversion failure (n = 64) | P-value |
|---|---|---|---|---|
| Age ≥ 65 years | 79 (44.6%) | 47 (41.6%) | 32 (50.0%) | 0.345 |
| Gender, male | 74 (41.8%) | 38 (33.6%) | 36 (56.3%) | 0.004 |
| BMI, kg/m$^2$ | 19.7 ± 3.0 | 19.9 ± 3.1 | 19.4 ± 2.8 | 0.301 |
| Smoking, pack-years | 34.6 ± 26.4 | 37.0 ± 32.4 | 32.6 ± 20.7 | 0.625 |
| Time from diagnosis to treatment, months | 10.6 ± 18.9 | 13.5 ± 22.2 | 5.4 ± 8.7 | 0.009 |
| Comorbidities | | | | |
| Hypertension | 51 (28.8%) | 33 (29.2%) | 18 (28.1%) | 1.000 |
| Diabetes | 24 (13.6%) | 18 (15.9%) | 6 (9.4%) | 0.260 |
| Chronic liver disease | 13 (7.3%) | 7 (6.2%) | 6 (9.4%) | 0.550 |
| Chronic kidney disease | 6 (3.4%) | 4 (3.5%) | 2 (3.1%) | 1.000 |
| Cardiovascular disease | 20 (11.3%) | 10 (8.8%) | 10 (15.6%) | 0.217 |
| Malignancy | 45 (25.4%) | 31 (27.4%) | 14 (21.9%) | 0.475 |
| Laboratory test | | | | |
| PNI[*] at time of diagnosis | 51.8 ± 9.2 | 52.9 ± 9.7 | 49.9 ± 6.6 | 0.033 |
| Malnutrition at time of Diagnosis (PNI[*] < 45) | 25 (14.1%) | 12 (10.6%) | 13 (20.3%) | 0.114 |
| PNI[*] at time of treatment initiation[‡] | 50.4 ± 5.7 | 50.7 ± 5.2 | 49.8 ± 6.6 | 0.351 |
| Malnutrition at time of treatment start[‡] (PNI < 45) | 28 (15.8%) | 14 (12.3%) | 14 (21.9%) | 0.079 |
| Cavitary lesion on computed tomography | 71 (40.1%) | 39 (34.%) | 32 (50.0%) | 0.055 |
| AFB smear | | | | 0.015 |
| Negative | 142 (80.2%) | 98 (86.7%) | 44 (68.8%) | |
| 1+ or 2+ | 24 (13.6%) | 10 (8.8%) | 14 (21.9%) | |
| 3+ or 4+ | 11 (6.2%) | 5 (4.4%) | 6 (9.4%) | |
| Symptoms | | | | |
| Cough | 75 (42.4%) | 45 (39.8%) | 30 (46.9%) | 0.429 |
| Sputum | 80 (45.2%) | 47 (41.6%) | 33 (51.6%) | 0.212 |
| Dyspnea | 20 (11.3%) | 12 (10.6%) | 8 (12.5%) | 0.806 |
| Hemoptysis | 39 (22.0%) | 23 (20.4%) | 16 (25.0%) | 0.572 |
| Fever | 10 (5.6%) | 8 (7.1%) | 2 (3.1%) | 0.333 |
| Weakness | 6 (3.4%) | 6 (5.3%) | 0 (0.0%) | 0.088 |
| Treatment medication | | | | |
| Macrolide | 173 (97.7%) | 110 (97.3%) | 63 (98.4%) | 1.000 |
| Rifampin | 174 (98.3%) | 111 (98.2%) | 63 (98.4%) | 1.000 |
| Ethambutol | 170 (96.0%) | 111 (98.2%) | 59 (92.2%) | 0.100 |
| Isoniazid | 3 (1.7%) | 3 (2.7%) | 0 (0.0%) | 0.554 |
| Fluoroquinolones | 6 (3.4%) | 2 (1.8%) | 4 (6.3%) | 0.191 |
| Aminoglycosides | 29 (16.4%) | 15 (13.3%) | 14 (21.9%) | 0.145 |
| Other | 4 (2.3%) | 1 (0.9%) | 3 (4.7%) | 0.135 |
| Initial number of medications | | | | 0.648 |
| ≤ 2 | 5 (2.8%) | 3 (2.7%) | 2 (3.1%) | |
| 3 | 142 (80.2%) | 93 (82.3%) | 49 (76.6%) | |
| ≥4 | 30 (16.9%) | 17 (15.0%) | 13 (20.3%) | |
| Duration of treatment, months | 20.5 ± 9.8 | 16.9 ± 4.8 | 24.0 ± 11.3 | <0.001 |

Data are presented as number (%) for categorical variables and median (range) or mean ± standard deviation (SD) for continuous variables

List of Abbreviations: PNI, prognostic nutritional index; BMI, body mass index; AFB, acid-fast bacilli

[*]PNI = 10 × serum albumin value (g/dL) + 0.005 × total lymphocyte count in the peripheral blood (/mm$^3$)

[§]Among the 193 patients who received treatment longer than 12 months, culture conversion data were available for 177 patients

[‡]Among the 193 patients who received treatment longer than 12 months, PNI data at the time of treatment initiation were available for 178 patients

## Variables related to all-cause mortality

Kaplan-Meier survival curves stratified by malnutrition and non-malnutrition groups at the time of diagnosis are shown in Fig 2. Patients in the malnutrition group exhibited a significantly higher mortality rate (P < 0.001) than those in the non-malnutrition group. The relationships between all-cause mortality and clinical parameters, including malnutrition, were evaluated in Table 5. Univariate analysis revealed that age ≥ 65 years, male gender, lower BMI, malnutrition at the time of diagnosis (P < 0.001), treatment shorter than 12 months, and history of diabetes, chronic kidney disease, cardiovascular disease, and cancer were correlated significantly with all-cause mortality. Comparison of the contributions of these indices by multivariate Cox proportional hazards analyses demonstrates that malnutrition (hazard ratio: 2.755, 95% CI: 1.610–4.475, P < 0.001), age ≥ 65 years, male gender, lower BMI, history of cancer, and positive AFB smear were significant risk factors for all-cause mortality.

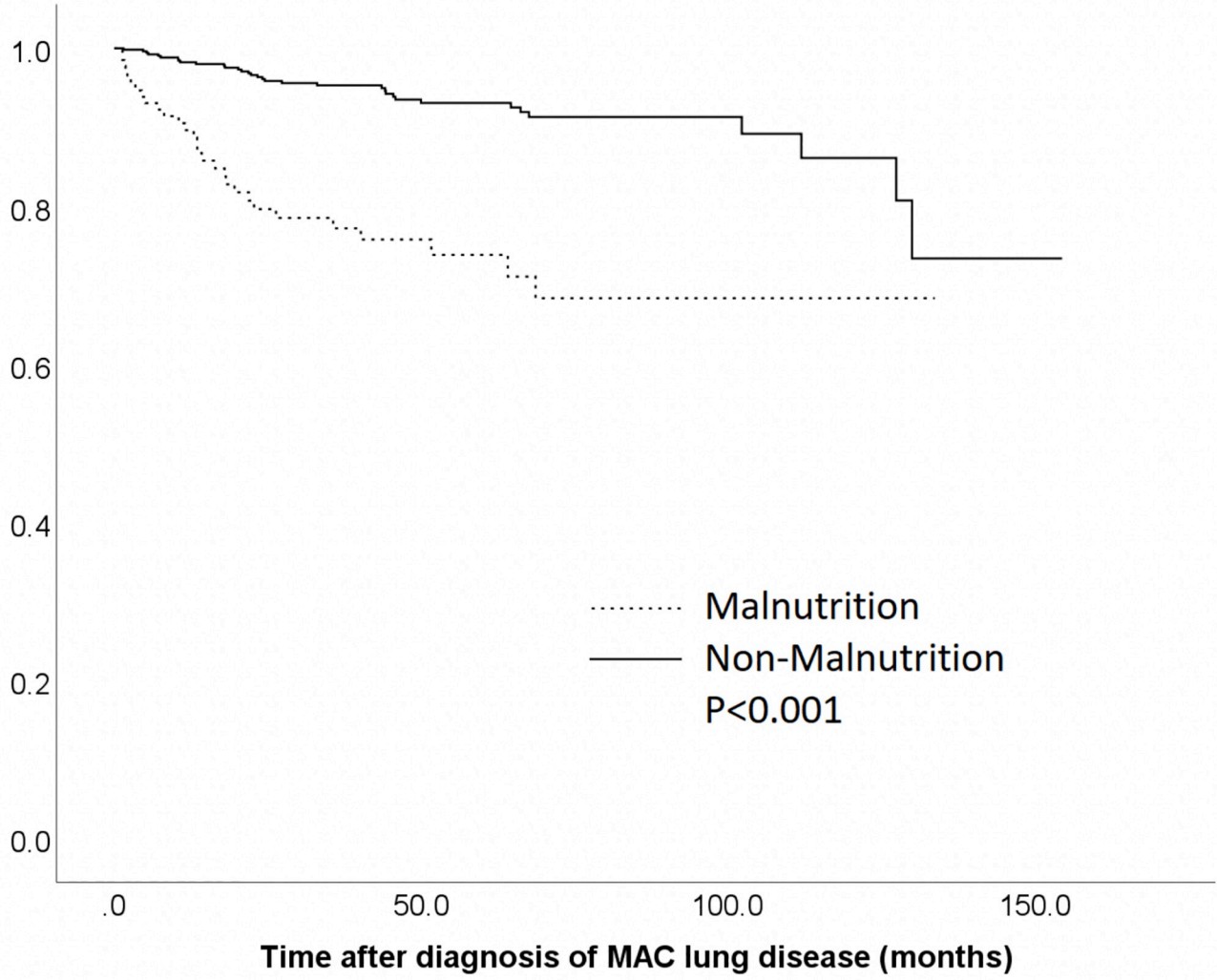

**Fig 2. Kaplan–Meier survival curves stratified by nutritional status.** Patients were divided into malnutrition and non-malnutrition groups according to prognostic nutritional index*. *Cut-off value of prognostic nutritional index for defining malnutrition and non-malnutrition was 45. List of Abbreviations; MAC, Mycobacterium avium complex.

**Table 5. Univariate and multivariate Cox's proportional hazard analyses of factors for mortality in Mycobacterium avium complex lung disease patients.**

| Variables | Among the deaths (n = 63) | Univariate | | Multivariate | |
|---|---|---|---|---|---|
| | | HR (95% CI) | P-value | HR (95% CI) | P-value |
| Age, years | 69.6 ± 10.7 | 1.060 (1.033–1.086) | <0.001 | | |
| Age ≥ 65 years | 46 (73.0%) | 2.960 (1.692–5.179) | <0.001 | 2.044 (1.075–3.887) | 0.029 |
| Sex, male | 46 (73.0%) | 3.792 (2.170–6.626) | <0.001 | 2.534 (1.376–4.664) | 0.003 |
| Body mass index, kg/m$^2$ | 19.0 ± 3.6 | 0.854 (0.786–0.927) | <0.001 | 0.858 (0.786–0.937) | 0.001 |
| Smoking, pack-years | 32.2 ± 19.3 | 1.006 (0.990–1.023) | 0.465 | | |
| NTM type, with cavity | 18 (28.6%) | 1.299 (0.751–2.248) | 0.349 | | |
| AFB smear | | | | | |
| Negative | 43 (68.3%) | Reference | | Reference | |
| 1+ or 2+ | 14 (22.2%) | 3.151 (1.705–5.822) | <0.001 | 2.849 (1.493–5.437) | 0.001 |
| 3+ or 4+ | 6 (9.5%) | 2.380 (0.981–5.775) | 0.055 | 2.477 (0.999–6.139) | 0.050 |
| Malnutrition, Diagnosis (PNI < 45) | 29 (46.0%) | 4.421 (2.686–7.277) | <0.001 | 2.755 (1.610–4.715) | <0.001 |
| PNI, diagnosis | 44.5 ± 10.4 | 0.906 (0.880–0.932) | <0.001 | | |
| Treatment started | 27 (42.9%) | 0.644 (0.388–1.071) | 0.090 | 0.496 (0.288–0.854) | 0.011 |
| Treatment longer than 12 months | 11 (17.5%) | 0.356 (0.185–0.686) | 0.002 | | |
| Treatment duration, months | 10.7 ± 8.2 | 0.941 (0.896–0.989) | 0.016 | | |
| Time from diagnosis to treatment, months | 6.9 ± 20 | 1.009 (0.979–1.039) | 0.576 | | |
| Hypertension | 30 (47.6%) | 1.560 (0.951–2.560) | 0.078 | 1.140 (0.629–2.067) | 0.665 |
| Diabetes | 16 (25.4%) | 1.950 (1.105–3.441) | 0.021 | 1.348 (0.723–2.514) | 0.347 |
| Chronic liver disease | 2 (3.2%) | 0.410 (0.100–1.678) | 0.215 | | |
| Chronic kidney disease | 6 (9.5%) | 2.209 (0.950–5.140) | 0.066 | 1.701 (0.685–4.227) | 0.253 |
| Cardiovascular disease | 16 (25.4%) | 1.987 (1.121–3.520) | 0.019 | 1.363 (0.718–2.590) | 0.344 |
| Cancer | 31 (49.2%) | 2.841 (1.731–4.664) | <0.001 | 2.477 (0.999–6.139) | 0.001 |

List of Abbreviations: HR, hazards ratio; 95% CI, 95% confidence interval; BMI, body mass index; PNI, prognostic nutritional index;

*PNI = 10 × serum albumin value (g/dL) + 0.005 × total lymphocyte count in the peripheral blood (1000 cells/μL)

## Discussion

Poor nutritional status represented by low BMI, low fat composition, and low level of serum albumin were reported as risk factors of progression of NTM PD. [25–27] Thus, experts recommend the treatment of MAC-PD when patients with considerable burden of disease (e.g., cavitary lesion on chest CT, AFB smear positive) have a poor nutritional status. However, the impact of poor nutritional status on the treatment outcome of MAC-PD has not been well evaluated.

To our knowledge, this is the first study to demonstrate the clinical utility of analyzing the relationship between PNI (as a composite value of malnutrition) and outcomes of MAC-PD. All-cause mortality was significantly higher in the malnutrition group than in the non-malnutrition group as defined by PNI at the time of diagnosis. Treatment intolerance, but not culture conversion rates after antibiotic therapy, was also related with malnutrition as defined by PNI at the time of treatment initiation. In addition, symptoms such as dyspnea, fever, and weakness were more common in patients with malnutrition than those in the non-malnutrition group. This may explain the relationship between malnutrition and treatment intolerance and mortality, as well as quality of life.

In this study, patients were divided into malnutrition and non-malnutrition groups according to the cut-off PNI score of 45, which has been reported to indicate moderate-to-severe malnutrition. [24] Conversely, in previous studies, the PNI cut-off value for defining malnutrition

varied between 40.0 and 50.0. [28–30] The optimal cut-off value and the division of the groups by PNI remain unclear. Further studies including prospective studies are needed to clarify the cut-off point not only for MAC-PD patients but for general non-surgical patients.

Numerous indicators have been used as markers to reflect nutritional status. Nutritional risk screening [5] uses information regarding food intake, BMI, and weight loss. Subjective global assessment [6] is a more detailed assessment that comprises patient history, as well as physical and subjective global assessment-specified variables. BMI is a well-known nutritional indicator; however, BMI distribution differs greatly across racial and ethnic populations. The nutritional status score [31] is calculated using serum cholesterol, lymphocyte, and albumin, and the Glasgow prognostic score [32] is calculated using albumin and C-reactive protein. In comparison, as mentioned before, PNI is calculated using only serum albumin levels and lymphocyte counts. PNI is therefore an easy and reliable tool with minor variability. [33]

PNI is known to reflect the nutritional and immune condition of patients. [23] Malnutrition reduces albumin concentration by decreasing its rate of synthesis; similarly, inflammation increases fractional catabolic rate, and in severe circumstances, it increases the permeability of vasculature, thereby allowing albumin to leak out into the extravascular space. [34] Furthermore, Siedner et al. reported that low serum albumin levels were strongly correlated with higher levels of the inflammatory marker interleukin 6, thus possibly indicating that low albumin levels might be a consequence of immune activation through mechanisms less directly associated with interleukin 6. [35] Malnutrition also results in lymphocyte deficiency. Chandra et al. [36] reported that patients with malnutrition had reduced lymphocyte counts. Lymphocytes are important to humoral immunity as well as cell-mediated immunity. [37, 38]

Malnutrition is reported to be risk factor of progression of NTM-PD, [25–27] and the outcome of mycobacterial infections is dependent on the interaction between the bacteria and the host's immune system. [39] Considering that PNI reflects both nutritional and immune status, low PNI scores could act as risk factors for the progression of MAC-PD. PNI scores may decrease because of advanced inflammation of MAC-PD and thus be related with higher mortality.

Treatment intolerance is particularly important in the treatment of MAC-PD. In the existing guideline, considerations on whether to start treatment, the age, baseline disease status, and risk benefit assessment of treatment effect and side effect are essential. In our study, one third of patients who initiated multiple antibiotics treatment discontinued the medication, as previously reported. [17] PNI was meaningful in predicting drug intolerance in our study. Further consideration of PNI may help predict drug intolerance and manage patients with MAC-PD.

Previous studies have demonstrated that nutritional interventions can improve a patient's lymphocyte [36] and albumin levels. [40] Nutritional support, therefore, may result in higher PNI scores. As our study demonstrated that higher PNI scores are related with better outcomes, nutritional interventions for MAC-PD might be helpful for patients with low PNI scores.

This study has some limitations. First, the study was retrospective and only included samples from a single center with limited number of patients and involved no replication cohort. Although multivariate analyses were performed, there are many compounding factors that influence the treatment and prognosis of MAC-PD. However, the validity of reported prognostic factors, such as BMI, and history of cancer or diabetes, was confirmed in this study population, thereby lending support to the present findings. Secondly, other methods of evaluating malnutrition other than BMI were not available at the time of our analysis. Integrating and comparing other indexes in future studies may help to further comprehend the nature and prognosis of MAC-PD. Thirdly, although PNI has been proposed as a simple surrogate

marker for evaluating immune-nutritional status, serum albumin levels and lymphocyte counts could also be influenced by a number of other factors, including medications, underlying disease, time, and age. [1]

## Conclusion

In the present study, we found that malnutrition, as defined by PNI, is a risk factor for all-cause mortality in MAC-PD patients. Treatment intolerance during antibiotic therapy was significantly higher in patients with malnutrition than in those in the non-malnutrition group. These findings suggest that the PNI score, which can be easily calculated using serum albumin levels and lymphocyte counts, is a useful prognostic marker for mortality and treatment intolerance. It is important to stress the importance of nutritional assessment for patients with MAC-PD.

## Supporting information

**S1 Table. Multivariate logistic regression analyses for variables related to culture conversion failure.**
(DOCX)

**S1 Dataset. Data set for MAC-PD study.**
(XLSX)

## Acknowledgments

The authors thank the patients and medical staff of Severance Hospital, Yonsei University. The authors are also grateful to Dr Kim Chi Young for assisting with data collection.

## Author Contributions

**Conceptualization:** Young Ae Kang.

**Data curation:** Sung Woo Moon, Eun Hye Lee, Ji Soo Choi, Ah Young Leem, Su Hwan Lee, Sang Hoon Lee, Song Yee Kim, Kyung Soo Chung, Ji Ye Jung, Moo Suk Park, Young Sam Kim, Young Ae Kang.

**Formal analysis:** Sung Woo Moon, Young Ae Kang.

**Investigation:** Sung Woo Moon, Young Ae Kang.

**Methodology:** Sung Woo Moon, Young Ae Kang.

**Supervision:** Young Ae Kang.

**Validation:** Young Ae Kang.

**Visualization:** Sung Woo Moon, Young Ae Kang.

**Writing – original draft:** Sung Woo Moon.

**Writing – review & editing:** Eun Hye Lee, Ah Young Leem, Su Hwan Lee, Sang Hoon Lee, Song Yee Kim, Kyung Soo Chung, Ji Ye Jung, Moo Suk Park, Young Sam Kim, Young Ae Kang.

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
