## [Decision Letter · Decision Letter 0]

11 Mar 2020

PONE-D-19-31183

Impact of prognostic nutritional index on outcomes in patients with Mycobacterium avium complex pulmonary disease

PLOS ONE

Dear Dr. Kang,

Thank you for submitting your manuscript to PLOS ONE. After careful consideration, we feel that it has merit but does not fully meet PLOS ONE’s publication criteria as it currently stands. Therefore, we invite you to submit a revised version of the manuscript that addresses the points raised during the review process.

The manuscript has been evaluated by three reviewers, and their comments are available below.

The reviewers have raised a number of concerns that need attention. They request additional information on methodological aspects of the study (such as more information on the characteristics of the populations compared in the analysis), and revisions to the statistical analyses. 

Could you please revise the manuscript to carefully address the concerns raised?

We would appreciate receiving your revised manuscript by Apr 24 2020 11:59PM. To enhance the reproducibility of your results, we recommend that if applicable you deposit your laboratory protocols in protocols.io, where a protocol can be assigned its own identifier (DOI) such that it can be cited independently in the future. For instructions see: http://journals.plos.org/plosone/s/submission-guidelines#loc-laboratory-protocols

We look forward to receiving your revised manuscript.

Kind regards,

Carmen Melatti

Associate Editor

PLOS ONE

Journal Requirements:

3. Thank you for including your ethics statement: We accessed and analyzed these data in compliance with the Health Insurance Portability and Accountability Act of 1996. This protocol was exempted from the institutional review board’s approval.

5. Please amend the manuscript submission data (via Edit Submission) to include author Kyung Soo Jung.

6. Please amend your authorship list in your manuscript file to include author Kyung Soo Chung.

Reviewers' comments:

Reviewer's Responses to Questions

**Comments to the Author**

1. Is the manuscript technically sound, and do the data support the conclusions?

Reviewer #1: Partly

Reviewer #2: Partly

Reviewer #3: Yes

2. Has the statistical analysis been performed appropriately and rigorously? 

Reviewer #1: No

Reviewer #2: Yes

Reviewer #3: Yes

3. Have the authors made all data underlying the findings in their manuscript fully available?

Reviewer #1: Yes

Reviewer #2: No

Reviewer #3: No

4. Is the manuscript presented in an intelligible fashion and written in standard English?

Reviewer #1: Yes

Reviewer #2: Yes

Reviewer #3: Yes

5. Review Comments to the Author

Reviewer #1: Authors reported useufullness of PNI in MAC-PD patients. Low PNI patients failed treatment and showed poor prognosis. Cut-off value of PNI should be calculated by ROC cureve or other methods. However, cut-off value of PNI was determined by other reports.

Reviewer #2: This study is a retrospective cohort analysis to evaluate whether patients with malnutrition, as defined by PNI, are high odds of intolerance to treatment, lower odds of achieving culture conversion after treatment, and higher risk of mortality than non-malnutrition patients.

I think this study is very interesting because this is the first study to analysis the relationship between PNI and outcome of MAC-PD.

I would like to request some revisions to make the article more sophisticated.

1, The authors concluded that malnutrition as defined by PNI is an independent predictor for treatment intolerance and all-cause mortality in patients with MAC-PD.

In table 1, there are many differences in the background between non-malnutrition group and malnutrition group. Although the authors performed a multivariate analysis, first of all, can malnutrition be an independent factor because there are so many compounding factors.

As you know, it is natural that patients with malnutrition are high age, easy to discontinue the treatment of MAC-PD, and therefore, they have poor prognosis.

The impact of the primary outcome in this study may be slightly weak.

2, As the disease of MAC-PD progresses, the nutrition status will get worsens.

I doubt that the patients with malnutrition group had more severe MAC-PD compared to non-malnutrition group. I think that the description regarding the status of the MAC-PD at diagnosis is poor.

Reviewer #3: This study evaluated the impact of PNI on outcomes in patients with MAC PD.

The manuscript was well written.

I have only a few comments.

Major comment

In Introduction section, please add the reason why you chose to evaluate PNI among the various nutrition assessment tools in this study.

Minor comment

page 12 line 11 "Based off" →　You mean "Based on" ？

6. PLOS authors have the option to publish the peer review history of their article (what does this mean?). If published, this will include your full peer review and any attached files.

Reviewer #1: No

Reviewer #2: No

Reviewer #3: No

---

## [Author Response · Author response to Decision Letter 0]

28 Mar 2020

Point by Point response to the editor

Reviewer #1

1. Authors reported useufullness of PNI in MAC-PD patients. Low PNI patients failed treatment and showed poor prognosis. Cut-off value of PNI should be calculated by ROC cureve or other methods. However, cut-off value of PNI was determined by other reports. 

Reply: Thank you for kind advice. Additionally, we tried receiver operating characteristic (ROC) curve for mortality to calculate the cut-off of PNI. 

The cut-off was 46.2 (Sensitivity 57.14, Specificity 77.53, AUC 0.698, P < 0.001). When using this cut-off, Multivariate logistic regression analyses for variables related to treatment intolerance (Table 3), Multivariate logistic regression analyses for variables related to culture conversion failure (S1 Table), and multivariate Cox’s proportional hazard analyses of factors for mortality in Mycobacterium avium complex lung disease patients (part of Table 5) changed as follows.

Table 3. Multivariate logistic regression analyses for variables related to treatment intolerance 

Variables OR (95% CI) P-value

Age ≥ 65 years 1.367 (0.950-1.969) 0.092

Gender, Male 0.847 (0.586-1.225) 0.377

Dyspnea 0.907 (0.515-1.598) 0.735

Hemoptysis 0.547 (0.353-0.847) 0.547

Malnutrition, treatment initiation (PNI <46.2) 1.866 (1.186-2.937) 0.007

S1 Table. Multivariate logistic regression analyses for variables related to culture conversion failure

Variables OR (95% CI) P-value

Age ≥ 65 years 1.135 (0.587-2.194) 0.707

Gender, Male 1.246 (0.628-2.474) 0.529

Cavitary lesion on computed tomography 1.415 (.0746-2.685) 0.288

Positive AFB Smear 2.350 (1.069-5.163) 0.033

Time from diagnosis to treatment, months 0.981 (0.961-1.002) 0.081

Malnutrition (PNI* <45) 1.236 (0.595-2.567) 0.571

Table 5. Univariate and multivariate Cox’s proportional hazard analyses of factors for mortality in Mycobacterium avium complex lung disease patients

Variables Multivariate

 HR (95% CI) P-value

Age ≥ 65 years 1.908 (0.998-3.649) 0.051

Sex, male 2.379 (1.289-4.393) 0.006

Body mass index, kg/m2 0.866 (0.793-0.946) 0.001

Positive AFB smear 2.838 (1.576-5.111) 0.001

Malnutrition, Diagnosis (PNI < 46.2) 2.812 (1.623-4.872) <0.001

Treatment started 0.559 (0.321-0.973) 0.040

Hypertension 1.208 (0.659-2.215) 0.542

Diabetes 1.356 (0.726-2.567) 0.350

Chronic kidney disease 1.709 (0.682-4.277) 0.253

Cardiovascular disease 1.218 (0.630-2.355) 0.558

Cancer 2.443 (1.410-4.233) 0.001

As we could see from the results above, that low PNI is a risk factor for mortality and treatment intolerance did not change and consistent reliability of PNI in MAC-PD was confirmed. But the problem of using ROC curve in defining cut-off is that through the paper, we wanted to analyze not only mortality but intolerance and treatment response and thought that using multiple cut-off could only cause confusion. Also, the cut-off of 45 was used as the cut-off 45 was not verified in MAC-PD patients. But we believe that through prospective study on MAC-PD patients, clarifying a new cut-off for malnutrition is needed. The sentence in discussion section has been changed. (Page 15, line 18-19)

Reviewer #2

This study is a retrospective cohort analysis to evaluate whether patients with malnutrition, as defined by PNI, are high odds of intolerance to treatment, lower odds of achieving culture conversion after treatment, and higher risk of mortality than non-malnutrition patients.

I think this study is very interesting because this is the first study to analysis the relationship between PNI and outcome of MAC-PD.

I would like to request some revisions to make the article more sophisticated. 

Reply: We appreciate your kind review.

1. The authors concluded that malnutrition as defined by PNI is an independent predictor for treatment intolerance and all-cause mortality in patients with MAC-PD.

In table 1, there are many differences in the background between non-malnutrition group and malnutrition group. Although the authors performed a multivariate analysis, first of all, can malnutrition be an independent factor because there are so many compounding factors.

As you know, it is natural that patients with malnutrition are high age, easy to discontinue the treatment of MAC-PD, and therefore, they have poor prognosis.

The impact of the primary outcome in this study may be slightly weak.

Reply: We totally agree with your valuable advice. But as the prevalence of MAC-PD is low and due to the retrospective nature of the study, we could not match variables nor add more variables due to limited number of patients. This paper was first written in 2017 but was delayed to additionally recruit patient data. We tried univariate Cox regression analysis first and then put the meaningful variables in the multivariate analysis in the analysis, as you mentioned, to identify whether malnutrition can be an independent factor excluding many compounding factors. And we also checked the multicollinearity between the variables. We tried using the term ‘related’ rather than ‘associated’ and the remaining term ‘associated’ was changed into ‘related’ and added the point the reviewer pointed in the limitation part. (Page 15, line 11 and Page 17, line 6-8)

2. As the disease of MAC-PD progresses, the nutrition status will get worsens.

I doubt that the patients with malnutrition group had more severe MAC-PD compared to non-malnutrition group. I think that the description regarding the status of the MAC-PD at diagnosis is poor.

Reply: Thank you for your wise advice. Until now, there is no authorized way to evaluate MAC-PD's severity, but currently, the presence of cavitary lesion on chest CT and sputum AFB smears are frequently used to examine the severity of MAC-PD in the practice. In our analysis, we included the presence of cavitary lesion on chest CT. And about the AFB smear, it was included in the analysis, but AFB smear was only categorized into positive or negative. To give additional data on the status of MAC-PD at diagnosis, the AFB status was additionally categorized into “negative”, “1+ and 2+”, and “3+ and 4+”. We wish this could give additional data on the severity of MAC-PD. Analyses were performed again and the numbers in the paper have changed accordingly. (Abstract, Result section, Tables, S1 Table) 

Reviewer #3

This study evaluated the impact of PNI on outcomes in patients with MAC PD.

The manuscript was well written.

I have only a few comments. 

Reply: We appreciate your kind review. 

Major

1. In Introduction section, please add the reason why you chose to evaluate PNI among the various nutrition assessment tools in this study.

Reply: Thank you for your valuable advice. We additionally added the reason why we chose to evaluate PNI among the various nutrition assessment tools from discussion to introduction section. (Page 3, line 6-7)

Minor

1. page 12 line 11 "Based off" →　You mean "Based on" ？ 

Reply: Thank you for your correction. We have changed “Based off” to “Based on”. (Page 6, line 14)

---

## [Editor Report · Decision Letter 1]

21 Apr 2020

Impact of prognostic nutritional index on outcomes in patients with Mycobacterium avium complex pulmonary disease

PONE-D-19-31183R1

Dear Dr. Young Ae Kang,

We are pleased to inform you that your manuscript has been judged scientifically suitable for publication and will be formally accepted for publication once it complies with all outstanding technical requirements.

With kind regards,

Abdelwahab Omri, Pharm B, Ph.D

Academic Editor

PLOS ONE

---

## [Editor Report · Acceptance letter]

24 Apr 2020

PONE-D-19-31183R1 

Impact of prognostic nutritional index on outcomes in patients with *Mycobacterium avium* complex pulmonary disease 

Dear Dr. Kang:

I am pleased to inform you that your manuscript has been deemed suitable for publication in PLOS ONE. Congratulations! Your manuscript is now with our production department. 

With kind regards,

on behalf of

Dr. Abdelwahab Omri 

Academic Editor

PLOS ONE